# Evaluation of Chronic Dietary Risk of Trifloxystrobin and Bupirimate in Cucumber Based on Supervised Residue Test

**DOI:** 10.3390/foods14101745

**Published:** 2025-05-14

**Authors:** Yanli Qi, Weirong Wang, Pengcheng Ren, Shu Qin, Jindong Li, Junli Cao

**Affiliations:** Shanxi Center for Testing of Functional Agro-Products, Shanxi Agricultural University, No. 79, Longcheng Street, Taiyuan 030031, China; qiyanli@sxau.edu.cn (Y.Q.); wangweirong93455@outlook.com (W.W.); renpengcheng@sxau.edu.cn (P.R.); qinshu@sxau.edu.cn (S.Q.); lijindong@sxau.edu.cn (J.L.)

**Keywords:** trifloxystrobin, bupirimate, metabolite, residue, dietary risk assessment

## Abstract

Trifloxystrobin and bupirimate are widely used as fungicides for controlling powdery mildew in cucumber cultivation. Supervised field trials were conducted in 12 representative regions across China, following Good Agricultural Practices (GAP) guidelines, to investigate their residue patterns and potential dietary exposure risks. Cucumber samples were analyzed using a validated method involving extraction with acidified acetonitrile (2% acetic acid, *v*/*v*), cleanup with primary secondary amine (PSA) and graphitised carbon black (GCB), and quantification by high-performance liquid chromatography-tandem mass spectrometry (HPLC-MS/MS). The method demonstrated excellent recovery rates (85–103%) throughout four spiking levels (0.01, 0.1, 0.3, and 1 mg/kg), with relative standard deviations (RSD) ≤ 4.8%. At 3 days after treatment, the residues of trifloxystrobin (including trifloxystrobin acid), bupirimate, and ethirimol in cucumbers were found to range from <0.01 to 0.013 mg/kg, <0.01 to 0.076 mg/kg, and <0.01 to 0.04 mg/kg, respectively. A chronic dietary risk assessment was conducted using a probabilistic model. The results showed an acceptable chronic risk (RQ_c_ ≤ 2.476%) for trifloxystrobin, bupirimate, and ethirimol across different sexes and ages, supporting the conclusion that the use of these fungicides in cucumber cultivation under the tested conditions was safe for Chinese consumers. More research was needed on children because they are at higher risk than other groups.

## 1. Introduction

Cucumbers (*Cucumis sativus* L.) are globally important cucurbit crops, widely consumed fresh or processed due to their nutritional profile, including vitamins (A, C, and K), minerals (potassium and magnesium), dietary fibre, and bioactive compounds such as cucurbitacins [1,2]. Recognized as a low-caloric vegetable with high water content, cucumbers significantly affect hydration and micronutrient intake across diverse populations. In China, cucumber production holds substantial agricultural and economic importance, with cultivation areas expanding to 1.31 million hectares in 2021 (https://www.fao.org/faostat/en/#data/QCL, accessed on 29 November 2024), positioning the nation as both a leading producer and consumer. A significant challenge in cucumber cultivation is powdery mildew [3], a highly destructive leaf disease caused by the obligate nutritional fungus *Podosphaera xanthii*. It reduces yield by 10% to 40%, compromising marketability by inhibiting photosynthesis and inducing epidermal damage. Regularly applying fungicides constitutes the most effective strategy for disease prevention and treatment [4]. However, the potential environmental impacts of pesticide residues necessitate a rigorous evaluation of their residual dynamics. Pesticide residues can pose risks to the environment and human health. On one hand, pesticide residues can contaminate soil and water resources through runoff and leaching, causing harm to non-target organisms and disrupting the ecological balance. On the other hand, long-term exposure to pesticide residues may pose chronic health risks to farm workers and consumers. Given pesticide residues’ potential environmental impacts, a rigorous evaluation of their residual dynamics is essential to ensure compliance with sustainable agricultural practices.

The fungicidal formulation 25% trifloxystrobin bupirimate emulsifiable concentrate (EC) has been granted global regulatory approval for combating cucumber powdery mildew. This combination enhances disease control effectiveness and slows resistance development by acting on separate molecular targets. Trifloxystrobin, (methyl(E)-methoxyimino-[1]acetate) belongs to the strobilurin fungicide, acts as a quinone outside inhibitor to inhibit the mitochondrial respiration of pathogens and has good efficacy against fungal diseases. Bupirimate, 5-butyl-2-(ethylaMino)-6-MethylpyriMidin-4-yldiMethylsulfaMate, a pyrimidine amine fungicide, inhibits fungal RNA polymerase III, blocking nucleic acid biosynthesis. Post-application metabolic transformations in plants yield bioactive derivatives: trifloxystrobin undergoes hydrolysis to trifloxystrobin acid [5], while bupirimate is demethylated to ethirimol (5-butyl-2-ethylamino-6-methylpyrimidin-4-ol) [6]. Notably, these metabolites exhibit elevated environmental persistence and toxicological profiles compared to parent compounds. Consequently, regulatory frameworks mandate comprehensive residue monitoring. The dietary risk assessment of trifloxystrobin encompasses the sum of trifloxystrobin and its acid metabolite [7]. According to GB 2763-2021 (National for safety standard-Maximum residue limits for pesticides in food) and “Pesticide Registration Residue Test List of Residues to be Tested and Dietary Risk Assessment Residues for Plant-derived Foods” [8], the dietary risk assessment of bupirimate residues is defined as bupirimate and ethirimol.

Residues of trifloxystrobin and bupirimate have been reported in various agricultural commodities, including cowpeas [9], apples [10], raisins, apricots [11], citrus fruits [12,13], chilli [14], pepper [15], rice [16], watermelon [17], and wheat [18,19]. Chen et al. employed ultra-high-performance liquid chromatography-tandem mass spectrometry (UHPLC-MS/MS) to quantify degradation kinetics and terminal residues of trifloxystrobin and its acid metabolite in cucumber, demonstrating first-order dissipation behaviour with half-lives of 2.1–3.5 days [20]. Bi et al. conducted comprehensive risk assessments in animal-derived products (milk, eggs, pork), concluding that residual concentrations (≤0.012 mg/kg) posed negligible dietary exposure risks [21]. However, a study on the residue behaviour of trifloxystrobin in tomatoes found that its residue levels may pose a potential risk to human health, with terminal residues (0.15–0.28 mg/kg) approaching the maximum residue limit (MRL, 0.3 mg/kg) [22]. These studies indicate the necessity for crop-specific residue monitoring. However, there are no reports on the simultaneous detection of trifloxystrobin, bupirimate, and their metabolites in cucumbers. In addition, there is a lack of research on the assessment of residue levels and dietary risks of trifloxystrobin and bupirimate in cucumbers based on multi-location residue trial results.

This study aims to establish a reliable analytical method to evaluate the residual behaviour of trifloxystrobin, topiramate, and its metabolites in cucumbers and conduct chronic dietary risk assessments in combination with dietary consumption data, providing a scientific basis for the safe use of pesticides. In summary, the objectives of this study were (1) to establish and validate a simple and effective analytical method for detecting trifloxystrobin, bupirimate, and their metabolites in cucumber; (2) to conduct field trials following GAP across 12 cucumber-producing regions across China, investigating the terminal residues of trifloxystrobin and bupirimate in cucumbers; (3) to assess the long-term dietary risk for the general Chinese population by integrating dietary consumption data. This research aims to provide a scientific basis for using 25% trifloxystrobin bupirimate microemulsion in cucumber production.

## 2. Materials and Methods

### 2.1. Chemicals and Reagents

The reference standards of trifloxystrobin (99.3% purity), bupirimate (84.7% purity), and ethirimol (99.81% purity) were procured from Dr Ehrenstorfer GmbH (Augsburg, Germany). Trifloxystrobin acid, with a high purity of 99.90%, was acquired from Beijing First Standard Technology Co., Ltd. (Beijing, China). Standard solutions of each compound at a concentration of 1000 mg/L were prepared in acetonitrile and remained stable for six months. An intermediate solution containing both compounds at a concentration of 10 mg/L was also prepared in acetonitrile, but this solution was stabilized for only one month. Both types of solutions are stored in the refrigerator at a temperature of 4 °C.

LC-MS grade acetonitrile and formic acid were purchased from Thermo Fisher Scientific (Shanghai, China), and LC-MS grade methanol was acquired from Merck KGaA (Darmstadt, Germany). Analytical grade sodium chloride (NaCl) and anhydrous magnesium sulphate (MgSO_4_) were procured from Beihua Fine-Chemicals Co., (Beijing, China). Solid-phase extraction sorbents, PSA (40 µm) and GCB (40 µm), were provided by Agela Technologies (Tianjin, China). 0.22 μm polytetrafluoroethylene (PTFE) syringe filter (Millex-FG, MilliporeSigma, Burlington, MA, USA). Ultra-pure water was purchased from Watsons Group (Guangzhou, China).

### 2.2. Field Experiments and Sampling

Field trials were conducted across 12 distinct geographical regions in China, comprising six open-field trials and six greenhouse trials; the site details were provided in Table 1. Each trial location included one treatment plot and one control plot, with each treatment plot covering 50 m^2^. For terminal residue analysis, a 25% trimoxystrobin·bupirimate microemulsion was applied twice at the maximum recommended dosage (105 g a.i./hm^2^), with a 7-day interval between applications. Based on the “Guideline for the testing of pesticide residues in crops (NY/T 788-2018)” [23], cucumber samples (*n* = 2 independent replicates) were collected 3, 5, and 7 days after the final application. Pretreatment blank samples were collected before pesticide application to establish baseline residue levels and mitigate potential cross-contamination. Following collection, cucumber samples were chopped, homogenized, and subsampled using the quartering method, then labelled and stored at ≤−18 °C fridge until analysis.

### 2.3. Extraction and Cleanup of Samples

Cucumber sample extraction was carried out using QuEChERS technique with some modifications [24]. Quechers technology consists of three steps, including acetonitrile extraction, salting out and dispersed solid phase extraction (DSPE) purification. Cucumber samples with an accurate weight of 10.00 ± 0.01 g was transferred to a 50 mL polypropylene centrifuge tube. Then, 10 mL of acidified acetonitrile (2% acetic acid, *v*/*v*) was added. The mixture was extracted using a multi-tube vortex mixer at 2500 rpm for 10 min. Following this, 4 g of anhydrous magnesium sulphate (MgSO_4_) and 1 g of sodium chloride (NaCl) were added, and the mixture was vigorously vortexed for 5 min. The sample was then centrifuged at 8000 rpm for 3 min.

Next, 1.5 mL of the organic supernatant was transferred to a 2 mL centrifuge tube containing 150 mg of MgSO_4_, 25 mg of PSA, and 25 mg of C18. The mixture was vortexed at 2500 rpm for 30 s, then centrifuged again at 8000 rpm for 3 min. Finally, the supernatant was passed through a 0.22 μm polytetrafluoroethylene (PTFE) membrane and injected into an injection vial for analysis by HPLC-MS/MS.

### 2.4. Instrument Analysis

The quantitative analysis of trifloxystrobin, trifloxystrobin acid, bupirimate, and ethirimol was conducted using a triple quadrupole mass spectrometer (SCIEX Triple Quad 4500, AB Sciex, Framingham, MA, USA) coupled with a high-performance liquid chromatography (HPLC) system. Chromatographic separation was achieved using an EVO C18 analytical column (50 mm × 2.1 mm i.d., 2.6 μm particle size; Phenomenex, Torrance, CA, USA) maintained at a temperature of 40 °C. The mobile phase consisted of (A) 0.1% (*v*/*v*) formic acid and 4 mM ammonium acetate in ultrapure water and (B) methanol (HPLC grade). A gradient elution programme was employed as follows: 0–0.5 min, 90% A; 0.5–2.0 min, linear gradient to 10% A; 2.0–3.5 min, hold at 10% A; 3.5–3.8 min, linear gradient back to 90% A; 3.8–5.0 min, re-equilibration at 90% A. The flow rate was maintained at 0.3 mL/min with an injection volume of 2 μL.

Mass spectrometric detection was performed using electrospray ionization (ESI) in positive ion mode with multiple reaction monitoring (MRM). The optimized instrument parameters were as follows: ion spray voltage at 5.5 kV, source temperature at 550 °C, curtain gas (nitrogen) pressure at 30 psi, and collision gas (nitrogen) pressure. The MRM transitions and the corresponding optimized parameters are detailed in Appendix A.

### 2.5. Method Validation and Quality Control

The analytical method was rigorously validated in compliance with European Commission guidelines (SANTE/11312/2021) for pesticide residue analysis [25]. The validation parameters included linearity, specificity, sensitivity (limit of quantification, LOQ), matrix effects (ME), accuracy, and precision.

Matrix-matched calibration standards were prepared at six concentrations (0.01, 0.02, 0.05, 0.1, 0.2 and 0.4 mg/L) by diluting a primary stock solution (10 mg/L in HPLC-grade acetonitrile) with extracts from blank cucumber samples. In parallel, solvent-based calibration standards were created using pure solvent for comparative analysis. The matrix effect (ME) was quantitatively assessed by comparing the slope ratios of the matrix-matched and solvent-based calibration curves, following the equation:(1)$$\;\mathrm{ME}=\frac{S_{matrix}-S_{\mathrm{solvent}}}{S_{\mathrm{solvent}}}\;\times 100\%\;$$
where S_matrix_ and S_solvent_ represent the slopes of the matrix-matched and solvent standard curves, respectively. |ME| < 20%: Negligible matrix effect; 20% ≤ |ME| ≤ 50%: Moderate matrix effect; |ME| > 50%: Strong matrix effect [26,27].

The accuracy and precision of the method were verified through quintuplicate recovery experiments conducted at four different spiking levels: 0.01, 0.1, 0.3, and 1 mg/kg in blank cucumber matrices. All analytes achieved satisfactory recovery rates ranging from 70% to 120%, with a relative standard deviation (RSD) of less than 15%. The limit of quantification (LOQ) was established as the lowest validated spiked level, which was 0.01 mg/kg, and provided signal-to-noise ratios greater than 10.

### 2.6. Chronic (Long-Term) Intake Risk Assessment

Risk assessments of pesticides should consider hazards and the possibility of pesticide exposure to humans. The study focused on the risks of dietary intake, which is expressed as risk quotient (RQ). The National Estimated Daily Intake (NEDI) and the chronic risk quotient (RQ_c_) were calculated using the method recommended by Joint Meeting on Pesticide Residues (JMPR). Deterministic and probabilistic models were used for risk assessment for various populations in China. The chronic risks were calculated for different ages (0–3, 3–5, 6–14, 15–49, 50–74 and ≥75 years old) and sex groups (male and female). Generally, the larger the HQ_c_ values are, the higher the dietary intake risks are; a value larger than 1 represents an unacceptable risk. The calculation method was as follows:(2)$$\mathrm{NEDI}=\frac{\sum (\mathrm{STMR}\times \mathrm{Fi})}{\mathrm{BW}}$$(3)$${\mathrm{RQ}}_{c}=\frac{\mathrm{NEDI}}{\mathrm{ADI}}$$
where STMR refers to the supervised trials median residue (mg/kg), Fi indicates dietary consumption (kg/d), and ADI represents the acceptable daily intake (mg/kg bw), and were obtained from the GB 2763-2021 [8]. BW stands for the average body weight (kg).

### 2.7. Statistical Analysis

MassLynx v4.2 software (Waters Corp, Milford, MA, USA) was utilized to obtain analysis data on the residual contents of trifloxystrobin, bupirimate, trifloxystrobin acid, and ethirimol. When the concentrations of the detected substances were below the LOQ, they were recorded as zero. The residual distribution heat map of trifloxystrobin and bupirimate and the risk assessment heat map for different populations were generated using the “pheatmap” package in R (version 4.4.2).

## 3. Results and Discussion

### 3.1. Validation of Analytical Methods

The potential threat to human health of pesticide residues in agricultural products has attracted widespread attention from consumers. The application of reliable sample pretreatment technology to extract is an important step in achieving pesticide residues in food. The QuEChERS method has been widely used to analyze agricultural and veterinary drug residues in food, especially fruits and vegetables with high water content samples [24]. In this study, the modified QuEChERS pretreatment combined with high-performance liquid chromatography-tandem mass spectrometry achieved quantitative monitoring of trifloxystrobin, trifloxystrobin acid, bupirimate and ethirimol in cucumbers.

Standard working solutions of trifloxystrobin, bupirimate, trifloxystrobin acid, and ethirimol were prepared at concentrations of 0.01, 0.02, 0.05, 0.1, 0.2, and 0.4 mg/L by dilution with blank cucumber matrix extract and pure solvent. Blank cucumber samples and solvent were extracted and analyzed according to the established method, and no background interference was found at the retention time of all analytes. Satisfactory linearity was obtained for all analytes with correlation coefficients (R^2^) ≥ 0.9900 (Table 2). The ME caused by ionization competition between co-eluting matrix components and target compounds was evaluated to ensure analytical accuracy [28]. It is crucial to determine the matrix effect when developing an analytical method, particularly in HPLC-MS/MS analysis utilizing an electrospray ionization source [29]. In this study, ME was typically evaluated by comparing the slopes of the matrix-matched standard curve with those of the solvent standard curve. The ME values for various analytes in the cucumber matrix ranged from −39.46% to −22.57% (Table 2), indicating significant ion suppression. Consequently, all analytes were quantified based on a matrix-matched calibration curve to mitigate the potential impact of co-extractables.

The analytical method employed in the study has been thoroughly validated to ensure both accuracy and precision, which are critical for reliable quantification of the target compounds in complex matrices like cucumber tissue. Systematic validation was carried out via spiked recovery experiments, a gold standard approach for assessing method performance in analytical chemistry. As indicated in Table 3, the average spiked recovery rates for the target compounds ranged from 85% to 103%. The relative standard deviation (RSD) for all compounds was tightly controlled and remained below 4.8%, fully complying with the repeatability limits set out in the European Commission guidelines (SANTE/11312/2021) [25]. The limit of quantification (LOQ) for all analytes in cucumber was determined to be 0.01 mg/kg.

### 3.2. Terminal Residues of Trifloxystrobin in Cucumbers

Under the Code of GAP for pesticide application, two independent composite samples (whole cucumbers) were collected at each test site during each pre-harvest interval (PHI). Figure 1 and Appendix A showed the final residue levels of trifloxystrobin, bupirimate, and their metabolites in cucumbers. The residue levels of trifloxystrobin decreased from below the limit of quantification (LOQ) of 0.010 mg/kg to 0.040 mg/kg at 3 days, and then to below LOQ to 0.022 mg/kg at 5 days, ultimately reaching below LOQ levels by 7 days. For bupirimate, residues ranged from below 0.010 mg/kg to 0.076 mg/kg at 3 days, below 0.010 mg/kg to 0.011 mg/kg at 5 days, and below 0.010 mg/kg at 7 days. Meanwhile, ethirimol residues were below 0.010 mg/kg to 0.030 mg/kg at 3 days, below 0.010 mg/kg to 0.16 mg/kg at 5 days, and below 0.010 mg/kg to 0.036 mg/kg at 7 days. These results indicate that trifloxystrobin and bupirimate were rapidly metabolized to trifloxystrobin acid and ethirimol, consistent with previous studies [30]. The maximum residue limits (MRL) for trifloxystrobin set by the European Union, the United States, China, South Korea, Japan, and Australia are 0.3, 0.5, 0.3, 0.5, 0.7, and 0.1 mg/kg, respectively. Regarding bupirimate on cucumbers, the European Union, China, and Japan have established MRL of 2, 0.5, and 1 mg/kg, respectively. As for ethirimol on cucumbers, the European Union and China have stipulated MRL of 0.2 and 1 mg/kg, respectively. The final residue levels of trifloxystrobin (including trifloxystrobin acid), bupirimate, and ethirimol were all found to be below their respective MRL.

### 3.3. Chronic Dietary Intake Rrisk Assessment

The dietary risk assessment of pesticide residues in agricultural commodities involves systematically evaluating the probability and severity of adverse health effects of chronic exposure to these chemical contaminants. This scientific process integrates data on toxicological profiles and residue dynamics while considering population-specific dietary consumption patterns [31]. The Risk Quotient (RQ_c_) is the primary metric for assessing potential health risks associated with long-term exposure. The RQ_c_ is calculated through a comparative analysis of the National Estimated Daily Intake (NEDI) against the product of average body weight (bw) and the acceptable daily intake (ADI) [32,33]. The NEDI calculation comprises two essential components: (1) food consumption data derived from the National Nutrition and Health Surveillance survey of Chinese residents and (2) residue estimates based on the STMR values obtained from chemical assessments. When STMR values are unavailable, conservative risk estimates are produced using the MRL stipulated, ensuring precautionary risk management. The hierarchy for selecting MRLs follows this sequence: primary consideration is given to Chinese national standards (GB 2763) [8], followed by international guidelines from the Codex Alimentarius Commission (CAC), and, when necessary, references from standards established in the United States, European Union, Australia, South Korea, and Japan. Risk assessment protocols vary among pesticide metabolites: Trifloxystrobin residues are quantified as the sum of both the parent compound and its acid metabolite, in line with recommendations from the FAO/WHO Joint Meeting on Pesticide Residues (JMPR). Meanwhile, bupirimate and its metabolite ethirimol are evaluated independently. Current ADI values specified in GB 2763-2021 are 0.040, 0.050, and 0.035 mg/kg bw for trifloxystrobin, bupirimate, and ethirimol, respectively [8].

A deterministic risk assessment model was employed to evaluate the long-term dietary exposure risks of trifloxystrobin (including its acid metabolite), bupirimate, and ethirimol for the Chinese population by consuming cucumbers and other registered crops. These results were presented in Appendix A and Figure 2A–C. Trifloxystrobin is currently registered in China for 27 agricultural commodities, which encompass cereal crops (such as rice, wheat, and corn), vegetables (including garlic, onion, tomato, eggplant, pepper, cucumber, bitter gourd, cowpea, and potato), fruits (such as citrus, apple, loquat, peach, jujube, cherry, grape, kiwifruit, strawberry, bayberry, litchi, mango, and banana), and speciality crops (like day lily and sunflower). Following risk maximization principles, we selected the highest maximum residue limit (MRL) within each food category to obtain a conservative estimate. For instance, the MRL for tomatoes (0.7 mg/kg) in dark-coloured vegetables was higher than the lower values in other vegetables (e.g., pepper at 0.5 mg/kg) for chronic risk assessment. Other registered pesticide options included bupirimate (cucumber, grape, and strawberry) and ethirimol (cucumber, apple, and strawberry). The National Estimated Daily Intake (NEDI) analysis revealed varying exposure contributions: trifloxystrobin (including its metabolite) contributed 2.52 mg (10.60% of the Acceptable Daily Intake, ADI), bupirimate contributed 3.15 mg (0.8% of ADI), and ethirimol contributed 2.205 mg (0.3% of ADI). The consumption of cucumbers contributed modestly to the total NEDI: 0.69% for trifloxystrobin, 7.43% for bupirimate, and 4.02% for ethirimol. The risk quotient (RQ) for bupirimate in registered crops was highest in fruits (0.725% of ADI) and lower in light-coloured vegetables (0.058%). For ethirimol, the RQ in registered crops was the highest in fruits (1.52% of ADI) and lower in light-coloured vegetables (0.612%). Notably, the risk from cucumber exposure to trifloxystrobin was minimal (0.07% of ADI). All calculated risk quotients remained significantly below the 100% ADI threshold across food categories, confirming that the dietary intake levels were acceptable.

The study revealed variations in RQ_c_ values for trifloxystrobin (including its acid metabolite), bupirimate, and ethirimol based on different demographic groups (Appendix A and Figure 2D–F). At the 50th percentile, the RQ_c_ ranges were as follows: trifloxystrobin (0.000–0.144%), bupirimate (0.000–0.115%), and ethirimol (0.000–0.164%). These values increased at the 97.5th percentile, reaching 0.403–2.167% for trifloxystrobin, 0.322–1.733% for bupirimate, and 0.460–2.476% for ethirimol. Age-related sensitivity patterns indicated a trend of decreasing risk with age. Preschool children (ages 0–3 years) exhibited a 1.64–6.16-fold higher susceptibility to exposure than seniors (those over 70 years). Although children’s dietary habits were similar to those of adults, their significantly lower body weight was identified as a key factor contributing to this difference in sensitivity [34]. In the gender-specific analysis, women aged 0–49 years showed elevated RQ_c_ values: trifloxystrobin (0.895–2.167%), bupirimate (0.716–1.733%), and ethirimol (1.023–2.476%), which significantly exceeded the corresponding ranges for men (0.829–1.476%). This difference was associated with the lower average body mass in premenopausal women. In contrast, postmenopausal women (over 50 years) displayed lower RQ_c_ ranges (0.403–0.67% for trifloxystrobin, 0.329–0.534% for bupirimate, and 0.469–0.763% for ethirimol) compared to similarly aged men (0.505–0.694%, 0.404–0.556%, and 0.577–0.763%, respectively). Importantly, all calculated RQ_c_ values remained well below the 100% threshold across the different demographic groups, suggesting negligible public health risks associated with chronic cucumber consumption under current exposure scenarios.

### 3.4. Limitations and Future Perspectives

The study made significant progress, and the long-term dietary risk assessment indicated minimal health risks from pesticide intake via cucumber consumption. However, the evaluation was based on theoretical assumptions and simplified models that only considered cucumber consumption as the exposure pathway. In reality, dietary patterns are diverse, and cumulative pesticide residues from multiple agricultural products may pose additional risks. Therefore, our comprehensive evaluation might not fully address multi-pesticide exposure risks across the entire diet. To overcome these limitations, future investigations should incorporate more comprehensive dietary datasets. Furthermore, extended follow-up studies are required to validate the accuracy and reliability of these findings. Such improvements will strengthen the scientific foundation for developing food safety regulations and optimizing agricultural practices.

## 4. Conclusions

This study developed a robust analytical method for determining trifloxystrobin, its metabolite trifloxystrobin acid, bupirimate, and ethirimol residues in cucumbers. The optimized QuEChERS-based extraction protocol utilizing acidified acetonitrile (2% acetic acid, *v*/*v*), followed by purification with PSA, GCB, and anhydrous MgSO_4_, demonstrated excellent recovery rates (70–120%) with RSD < 15% when coupled with HPLC-MS/MS analysis. Matrix-matched calibration effectively compensated for signal suppression effects, ensuring accurate quantification. Field trials conducted under GAP conditions showed that application of 25% trifloxystrobin bupirimate microemulsion at the maximum recommended dosage (105 g a.i./hm^2^, two applications with 7-day interval) resulted in terminal residues were <0.01 mg/kg—0.16 mg/kg, meeting the MRL requirements. The long-term dietary risk assessment of trifloxystrobin, bupirimate, and ethirimol intake through cucumber consumption suggests that it did not pose an unacceptable risk to the general health of the Chinese population. In particular, the assessment, which conservatively used MRL values instead of median residue values and approximate dietary group consumption instead of individual food consumption, indicated that the actual dietary intake risk was even lower. Therefore, it is recommended that 25% trifloxystrobin bupirimate microemulsion be applied on cucumbers, and it should not exceed two times, with a PHI of 3 days.

## Figures and Tables

**Figure 1 foods-14-01745-f001:**
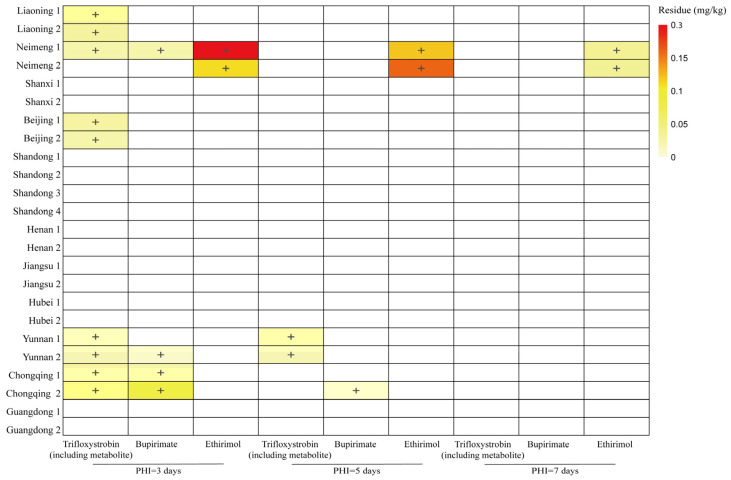
Colour map of pesticides detected in cucumber samples. +: Concentration > LOQ, the darker the colour, the greater the concentration.

**Figure 2 foods-14-01745-f002:**
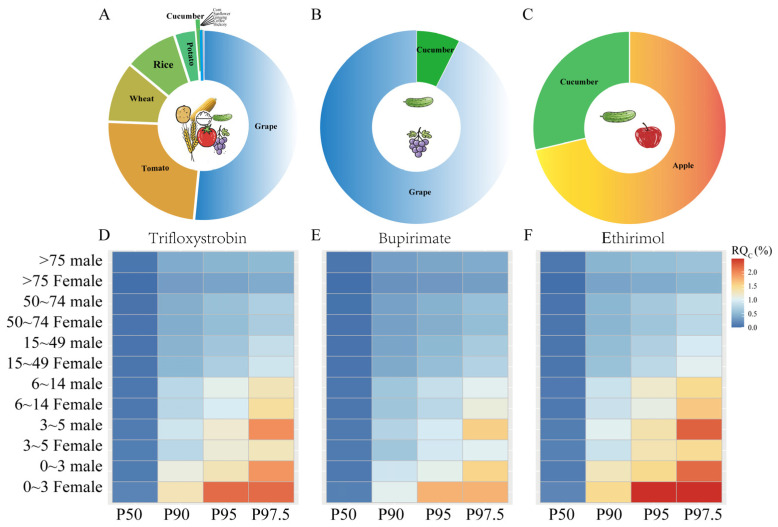
Chronic dietary risks of trifloxystrobin (including Trifloxystrobin acid), bupirimate and ethirimol (**A**–**C**). The long-term dietary risk of pesticides in all registered crops is determined deterministically (**D**–**F**), the probability estimate of the long-term dietary risk of pesticides in cucumbers. P50–P97.5 refers to different percentiles of the probability model.

**Table 1 foods-14-01745-t001:** The information on field trial.

Location (Province)	Cultivation	Soil Properties			Climate	
		pH	OMC (%)	CEC (cmol/kg)	Average Temperature (°C)	Rainfall (mm)
Shenyang (Liaoning province)	Greenhouse	6.9	3.1	19	24.2	0
Huhehaote (Neimeng province)	Greenhouse	8.0	3.9	12.5	20.5	0
Jinzhong (Shanxi province)	Field	8.4	1.7	21.2	25.82	23.6
Beijing	Greenhouse	7.9	1.2	13.8	/	0
Qingdao (Shandong Province)	Greenhouse	6.5	1.2	17	20.7	98
Taian (Shandong province)	Field	6.9	0.98	9.3	27.4	/
Xinxiang (Henan province)	Field	7.9	1.3	7.7	/	/
Nanjing (Jiangsu province)	Greenhouse	6.4	1.92	14.9	21.8	0.9
Lichuan (Hubei province)	Field	5.4	/	/	/	/
Kunming (Yunnan province)	Greenhouse	5.5	4.5	26.3	20.8	39
Chongqing	Field	6.5	2.3	18.5	20.9	41.5
Foshan (Guangdong province)	Field	5.9	1.02	3.9	28.5	131.2

/: Missing data; OMC: organic matter content; CEC: cation exchange capacity.

**Table 2 foods-14-01745-t002:** Regression parameters and matrix effects for trifloxystrobin, bupirimate, trifloxystrobin acid, and ethirimol on cucumbers.

Compound	Structural Formula	Matrix	Calibration Equation	Coefficient of Determination (R^2^)	Matrix Effects(ME, %)
Trifloxystrobin	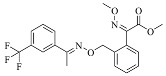	Solvent	y = 1.139 × 10^8^x + 7.02 × 10^5^	0.9900	-
Cucumber	y = 8.414 × 10^7^x + 8.46 × 10^4^	0.9998	−26.11%
Trifloxystrobin acid	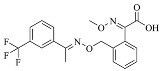	Solvent	y = 1.968 × 10^6^x + 3918	0.9977	-
Cucumber	y = 1.192 × 10^6^x − 2521	0.9948	−39.46%
Bupirimate	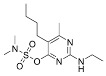	Solvent	y = 3.637 × 10^7^x − 23,820	0.9957	-
Cucumber	y = 2.554 × 10^7^x − 7.19 × 10^4^	0.9909	−29.77%
Ethirimol	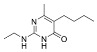	Solvent	y = 3.587 × 10^7^x − 7.24 × 10^4^	0.9939	-
Cucumber	y = 2.777 × 10^7^x − 8.51 × 10^4^	0.9905	−22.57%

**Table 3 foods-14-01745-t003:** Recoveries of trifloxystrobin, bupirimate, trifloxystrobin acid, and ethirimol in cucumbers.

Pesticides	0.01 mg/kg		0.1 mg/kg		0.3 mg/kg		1 mg/kg	
	Recoveries (%)	RSD (%)	Recoveries (%)	RSD (%)	Recoveries (%)	RSD (%)	Recoveries (%)	RSD (%)
Trifloxystrobin	101	1.2	97	1.3	100	1.6	103	4.0
Trifloxystrobin acid	85	4.7	91	2.3	86	2.6	97	4.8
Bupirimate	101	1.1	96	1.7	98	1.8	103	3.6
Ethirimol	102	3.2	93	2.3	97	2.9	98	3.2

## Data Availability

The original contributions presented in this study are included in the article/Appendix A. Further inquiries can be directed to the corresponding author.

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
