# Peer review of "Evaluation of Chronic Dietary Risk of Trifloxystrobin and Bupirimate in Cucumber Based on Supervised Residue Test"

_foods, 2025, doi:10.3390/foods14101745_

Round 1
Reviewer 1 Report
Comments and Suggestions for Authors
The article "Evaluation of Chronic Dietary Risk of Trifloxystrobin and Bupirimate in Cucumber Based on Supervised Residue Test", well written and clearly presented. It is of interest given the widespread use of both pesticides in protecting cucumbers from pests and diseases. I think this work is a very interesting line of research as it provides data on food safety. The analytical methodology is adequate and the ingestion risk analysis is appropriate, so I consider its publication appropriate.
However, I consider that reference should be made to the country to which the MRL with which one works corresponds (EU, USA, China; ...) (Line 232)
Author Response
|
Comments 1: The article "Evaluation of Chronic Dietary Risk of Trifloxystrobin and Bupirimate in Cucumber Based on Supervised Residue Test", well written and clearly presented. It is of interest given the widespread use of both pesticides in protecting cucumbers from pests and diseases. I think this work is a very interesting line of research as it provides data on food safety. The analytical methodology is adequate and the ingestion risk analysis is appropriate, so I consider its publication appropriate. |
|
Response 1: Dear reviewer, thank you for your positive feedback and affirmation of our manuscript. We are pleased that you found the article well-written, clearly presented, and of interest given the widespread use of the pesticides in question. We appreciate your acknowledgment of the study's contribution to food safety data and the appropriateness of our analytical methodology and risk analysis. We are glad to hear that you consider the work suitable for publication. |
|
Comments 2: However, I consider that reference should be made to the country to which the MRL with which one works corresponds (EU, USA, China; ...) (Line 232) |
|
Response 2: Dear reviewer, the maximum residue limits (MRL) for trifloxystrobin set by the European Union, the United States, China, South Korea, Japan, and Australia are 0.3, 0.5, 0.3, 0.5, 0.7, and 0.1 mg/kg respectively. Regarding bupirimate on cucumbers, the European Union, China, and Japan have established MRL of 2, 0.5, and 1 mg/kg respectively. As for ethirimol on cucumbers, the European Union and China have stipulated MRL of 0.2 and 1 mg/kg respectively. The final residue levels of trifloxystrobin (including trifloxystrobin acid), bupirimate, and ethirimol were all found to be below their respective MRL. These details can be found at lines 266–272. |
Reviewer 2 Report
Comments and Suggestions for Authors
The validation procedure for the analytical method using cucumber was found to be well aligned with international guidelines, and the risk assessment scenario was also well-constructed. The results of this study provide assurance of consumer safety and are expected to serve as valuable data for public health protection.
Author Response
Comments 1: The validation procedure for the analytical method using cucumber was found to be well aligned with international guidelines, and the risk assessment scenario was also well-constructed. The results of this study provide assurance of consumer safety and are expected to serve as valuable data for public health protection.
Response 1: Dear reviewer, we are deeply appreciative of your positive recognition regarding our manuscript.
Reviewer 3 Report
Comments and Suggestions for Authors
May 1, 2025
Dear authors,
After reviewing the article titled “Evaluation of Chronic Dietary Risk of Trifloxystrobin and Bupirimate in Cucumber Based on Supervised Residue Test” which was submitted for possible publication in the Foods journal. I consider that the work addresses an interesting and ad hoc topic with the scope of the journal. However, I think that some aspects could help it improve.
Abstract
The abstract is well structured, as it presents a brief and concise description of the study, including the research problem, objectives, methodology used, the most important results and the main conclusions.
Introduction
Line 39. The scientific nomenclature of the species Podosphaera xanthii must be placed in italics.
Line 45 denotes the percentage of each active ingredient in the formulation.
It is imperative that a reference is added to line 52. It is recommended that the following reference be appended for the purpose of providing a comprehensive overview of the modes of action of various pesticides.
https://irac-online.org/mode-of-action/classification-online/
Line 79, cyprodinil or bupirimate?
Materials and Methods
Line 100, -18ºC or 18ºC?
Lines 110-120, It is imperative to ascertain whether the samples taken were composite samples. To proceed with the analysis, it is necessary to determine which sampling guide was followed. It is recommended that the following sources be consulted for further information:
- https://www.gob.mx/senasica/documentos/manual-tecnico-de-muestreo-de-vegetales-para-la-determinacion-de-residuos-de-plaguicidas?state=published
- CXG-33-1999: Recommended Methods of Sampling for the Determination of Pesticide Residues for Compliance with MRLs (https://www.fao.org/fao-who-codexalimentarius/codex-texts/guidelines/en/)
Lines 122-132, It is imperative that references are added to the methodology that was followed.
Line 161, It is imperative that all concentrations present within the matrix match are indicated on the calibration curve.
Line 179, Do not use abbreviations without first stating their meaning at least once. For example, Joint Meeting on Pesticide Residues (JMPR). Apply to the whole document.
Since determinations are made at trace levels, it is incumbent upon the authors to employ Quality Assurance of the Method. It is imperative to incorporate a new subsection within the Materials and Methods section, wherein this critical information is delineated.
It is incumbent upon authors to indicate whether statistical analyses have been performed.
Results and discussion
Lines 216-217 It is imperative that the LOQ is expressed in milligrams per kilogram, given that the subject is a solid plant matrix.
In table 3 add the standard deviations in each of the recoveries shown at the different fortification levels.
I consider that a subsection called limitations, and future perspectives should be added to the results and discussion section.
Conclusions
The conclusions of the study are consistent with its objectives and are derived from the results. They are presented in a straightforward and accessible manner.
Author Response
|
Comments 1: Abstract:The abstract is well structured, as it presents a brief and concise description of the study, including the research problem, objectives, methodology used, the most important results and the main conclusions. |
|
Response 1: Thank you very much for your help with our manuscript. |
|
Comments 2: Line 39. The scientific nomenclature of the species Podosphaera xanthii must be placed in italics. |
|
Response 2: The name of the species has been changed to italics in line 39. |
|
Comments 3: Line 45 denotes the percentage of each active ingredient in the formulation. |
|
Response 3: Dear reviewer, the mixture is 25% trifloxystrobin·bupirimate emulsifiable concentrate (EC), in which the effective components of trifloxystrobin and bupirimate are 10% and 15% respectively. This change can be found in line 51. |
|
Comments 4: It is imperative that a reference is added to line 52. It is recommended that the following reference be appended for the purpose of providing a comprehensive overview of the modes of action of various pesticides. https://irac-online.org/mode-of-action/classification-online/ |
|
Response 4: Dear Reviewer, thank you kindly for your valuable assistance. I have meticulously examined the entirety of the website you referred to, which delves into the action mechanisms of a multitude of pesticides, such as acetylcholinesterase (ACHE) inhibitors, chloride channel blockers, sodium channel regulators, and others. Regrettably, this website falls short in providing information regarding the action mechanisms of trifloxystrobin and bupirimate. To fill this gap, we delved into the relevant literature to determine the action mechanisms of these two fungicides. Trifloxystrobin, a methoxyacrylate bactericide, effectively inhibits mitochondrial respiratory function by precisely blocking the electron transfer at the cytochrome bc1Qo site. On the other hand, bupirimate, a pyrimidine bactericide, exerts its bactericidal effect by inhibiting adenine nucleoside deaminase. This change can be found in line 54-59. |
|
Comments 5: Line 79, cyprodinil or bupirimate? |
|
Response 5: Dear reviewer, thank you for your careful inspection. This should be bupirimate, and the change can be found in line 86. |
|
Comments 6: Line 100, -18ºC or 18ºC? |
|
Response 6: Dear reviewers, the pesticide standard solution is stored in refrigeration at 4℃. This change can be found in line 109. |
|
Comments 7:Lines 110-120, It is imperative to ascertain whether the samples taken were composite samples. To proceed with the analysis, it is necessary to determine which sampling guide was followed. It is recommended that the following sources be consulted for further information: https://www.gob.mx/senasica/documentos/manual-tecnico-de-muestreo-de-vegetales-para-la-determinacion-de-residuos-de-plaguicidas?state=published CXG-33-1999: Recommended Methods of Sampling for the Determination of Pesticide Residues for Compliance with MRLs (https://www.fao.org/fao-who-codexalimentarius/codex-texts/guidelines/en/) |
|
Response 7: Dear reviewer, I accept your suggestion. When collecting samples in the field, we followed NY/T 788-2018 "Guideline for the testing of pesticide residues in crops" issued by the Ministry of Agriculture and Rural Affairs of the People's Republic of China, and the collected samples were independent parallel samples in the treatment plot. The two cucumber samples collected were from the same treatment cell and were duplicate samples (n=2). We have added guidelines to the manuscript, and the changes can be found in lines 124 -127. |
|
Comments 8: Lines 122-132, It is imperative that references are added to the methodology that was followed. |
|
Response 8: Reply 8: Dear reviewer, the extraction and purification methods of trifloxystrobin and bupirimate in cucumber follow the QuEChERS pretreatment method in the previously published paper, with some modifications. These changes can be found in lines 132-135. |
|
Comments 9: Line 161, It is imperative that all concentrations present within the matrix match are indicated on the calibration curve. |
|
Response 9: Dear reviewers, the matrix matching standard curve and solvent standard curve are determined by six concentrations, including 0.01, 0.02, 0.05, 0.1, 0.2 and 0.4 mg/L of all analysis. |
|
Comments 10: Line 179, Do not use abbreviations without first stating their meaning at least once. For example, Joint Meeting on Pesticide Residues (JMPR). Apply to the whole document. |
|
Response 10: Dear reviewers, I am happy to receive your comments. When it first appeared, JMPR should not use abbreviations. The changes can be found in lines 192 -193. |
|
Comments 11: Since determinations are made at trace levels, it is incumbent upon the authors to employ Quality Assurance of the Method. It is imperative to incorporate a new subsection within the Materials and Methods section, wherein this critical information is delineated. |
|
Response 11: Dear reviewer, we have changed the "2.5 Method validation" to "2.5 Method validation and quality control". Quality assurance is achieved through spiking recovery tests, by adding pesticide standards at a specific concentration to the blank matrix, the accuracy of the method is finally ensured through recovery. The changes can be found in lines 166. |
|
Comments 12: It is incumbent upon authors to indicate whether statistical analyses have been performed. |
|
Response 12: Dear reviewer, I accept your suggestions. I have added "2.7 Statistical Analysis" to the manuscript,please refer to Line 203-209. |
|
Comments 13: Lines 216-217. It is imperative that the LOQ is expressed in milligrams per kilogram, given that the subject is a solid plant matrix. |
|
Response 13: Reviewer, your suggestion is correct. We made a revision in the manuscript, and the LOQ was 0.01 mg/kg. Please refer to Line 247. |
|
Comments 14: In table 3 add the standard deviations in each of the recoveries shown at the different fortification levels. |
|
Response 14: Dear reviewer, we conducted spiked recovery trials at four levels, with each level repeated five times. Table 3 presents the recovery rates along with the relative standard deviations (RSD). The standard deviations you requested are indeed reflected in the RSD values provided in Table 3. |
|
Comments 15: I consider that a subsection called limitations, and future perspectives should be added to the results and discussion section. |
|
Response 15: Dear reviewer, we have incorporated your suggestion by adding a subsection on "Limitations and Future Perspectives" in the Results and Discussion section. We acknowledge that the chronic dietary risk assessment relies on theoretical assumptions and simplified models, focusing solely on cucumber consumption as the exposure route. In reality, dietary patterns are more complex, and cumulative pesticide residues from various agricultural products could contribute to additional risks. These limitations are detailed in lines 349-360 of the manuscript. |
|
Comments 16: The conclusions of the study are consistent with its objectives and are derived from the results. They are presented in a straightforward and accessible manner. |
|
Response 16: Thank you for your approval of our manuscript. |